# Female Medical Students’ Experiences of Sexism during Clinical Placements: A Qualitative Study

**DOI:** 10.3390/healthcare11071002

**Published:** 2023-03-31

**Authors:** Darya Ibrahim, Ruth Riley

**Affiliations:** 1Medical School, College of Medical and Dental Sciences, University of Birmingham, Birmingham B15 2TT, UK; darya.ibrahim@nhs.net; 2School of Health Sciences, Faculty of Health and Medical Sciences, University of Surrey, Guildford GU2 7YH, UK

**Keywords:** female, medical school, sex discrimination, sexism, sexual harassment, medical students, gender discrimination, discrimination, woman, medical education

## Abstract

In the UK, more women are studying medicine than men, most of whom have experienced sexism, yet these experiences are under-researched. This qualitative study explores female medical students’ experiences of sexism on placement, impacts sustained, barriers and facilitators encountered upon reporting. A total of 17 semi-structured interviews were conducted, employing purposive sampling, snowball sampling and an inductive thematic analysis. A qualitative methodology was underpinned by the feminist social constructionist theory. Four themes were identified: 1—experiences of sexism, comprising physical and verbal harassment and microaggressions; 2—negative impacts of sexist encounters ranged from psychosocial to repercussions on learning and development; 3—systemic and attitudinal barriers to reporting; 4—recommendations to tackle sexism shaped by the views and experiences of female medical student participants. Female medical students experienced wide-ranging sexism which negatively impacted their wellbeing with negative repercussions for their training and development. The barriers to reporting need to be urgently addressed, and systems, policies and processes need to be over-hauled to sensitively, effectively and equitably manage and provide justice to students who experience and report sexism. Students need to be empowered to respond, report and be offered psychological safety in doing so. Attitudes and practices which are complicit in sustaining sexism need to be challenged and changed.

## 1. Introduction

Sexism is defined as a form of discrimination or prejudice based on an individual’s sex and includes: “Any act, gesture, visual representation, spoken or written words, practice, or behaviour based upon the idea that a person or a group of persons is inferior because of their sex, which occurs in the public or private sphere, whether online or offline.” [1,2]. Sexual harassment is a sub-category of both sex discrimination and harassment. Such oppression stems from a variety of factors including a perpetrator’s beliefs, societal behaviours, culture, and oppressive social and institutional structures. It is defined as non-consensual behaviour of a sexual nature imposed upon an individual and can take the form of verbal, non-verbal or physical acts [3,4]. 

The United Nations Development Programme Report found that almost 90% of the global population surveyed were biased against women [5]. It has been estimated that 50–60% of women in the European Union and 30% of women in the United Kingdom (UK) have experienced sexual harassment in the workplace [6,7,8]. Attempts to curb sexism, including the formation of Gender Equality Policies and the Equality Act of 2010, have been largely ineffective [9,10]. 

Evidence suggests that within medicine, sexism is even more prevalent. According to a 2021 report by the British Medical Association (BMA), 91% of qualified female doctors have experienced sexism at work and 56% have encountered sexual harassment [11]. Figure 1 demonstrates the medical training pathway in England [12]. The victims ranged across all specialties and hierarchies [11,13]. Of the 2458 participants, 70% had felt their medical expertise challenged or not trusted due to their sex [11]. Of the participants who had encountered or witnessed sexism, 42% of participants did not feel supported in reporting [14]. These findings are consistent with previous research, which found that up to 70% of female staff in academic medicine reported gender-based discrimination, and 48% of female physicians reported sexist comments and 30% reported severe harassment [15]. Sexism and sexual misconduct in healthcare are becoming more openly discussed topics of conversation with multiple social movements such as the “#HeforShe” campaign and “#MeToo” as well as victims speaking out on the platform “ScrubSuvivors” [16,17,18,19].

There are 35 medical schools registered under the General Medical Council in the UK [20]. This study took place at the second largest medical school in the country, comprising of 2003 students, of which 64% are female [21,22]. This is in line with the national student average, which stands at 61% [23]. Despite being the largest student demographic, the proportions of females decrease with career progression, dropping to just 36% by consultancy level [24]. This has been largely attributed to sex discrimination and sexual harassment [14,25,26]. Discrimination has been identified as a key barrier to career progression and a negative influence on the mental and physical health of female employees [15,27,28,29].

Research with a specific focus on medical students’ experiences of sexism has been limited, although what little evidence is available suggests the aforementioned experiences are reflective of those of female medical students. Despite female medical students being over-represented in medical schools nationwide, available evidence shows that females continue to be discriminated against throughout their medical placements before they qualify [30]. Although quantitative research has shown that sexism still exists, throughout all medical grades, such a methodology does not sufficiently explore the experiences and capture their impact [14,20,21,27]. The purpose of this study is to expand on the limited qualitative research available. This study, employing a qualitative methodology, aims to provide an understanding of the challenges faced by female medical students in England and the way in which they navigate and negotiate such experiences. This entails highlighting the experiences and impacts, identifying barriers to reporting, and making recommendations to address sexism on placement. An additional objective is to provide a voice to empower the victims and raise awareness of this issue. Examining these issues in depth will inform the development of strategies to address these challenges and promote equality in medicine [31].

For the purpose of this study, the term female includes all individuals that self-identify within this category including those that classify themselves as women and non-binary individuals.

## 2. Methods

### 2.1. Study Design and Setting

Qualitative methods were used to explore experiences of sexism in female medical students. This paper was analysed through the sensitising lens of the Feminist Social Constructionist Theory (FSCT) as this best encapsulates such experiences [32]. The methodology was orientated using the FSCT in order to address critical knowledge gaps which perpetuate inequality in the lives of women and to mobilise research evidence to effect social change [33]. FSCT enables the use of an inclusive and empowering research design, attending to power through critically reflexive and inclusive practices and methods incorporating qualitative methods and female researchers [32,33]. Such an approach allowed for the study to sensitively attend to research ethics as well as participants’ wellbeing by validating their experiences and providing women with a voice. A feminist approach was used in order to not only produce research about women but also to provide research for women, allowing for the necessary people to be equipped with evidence to bring forth change [34]. 

This study took place at one of the largest medical schools in England with students attending a variety of clinical placements throughout the West Midlands region, ranging from large tertiary centres to smaller district general hospitals and primary care placements.

### 2.2. Recruitment and Sampling

Participants were recruited through a variety of methods, using advertisements in social media, a medical student email bulletin, hospital noticeboards and word of mouth. Purposive sampling was used to ensure maximum variation based on the following characteristics: gender, ethnicity, disability, year group, previous degrees, religion and sexual orientation. Potential participants that had shown interest were sent a Participant Information Leaflet (PIL). All participants were required to meet the inclusion criteria (Table 1). All 17 participants that were recruited were eligible and accepted to take part in the study. Participants consent was sought prior to being interviewed by completing a consent form and further confirmed at the start of the interview. Recruitment continued throughout the interview phase as snowball sampling was incorporated. One participant was recruited through such sampling technique. Online video interviews were booked at times suitable to the participants. Due to the potential for distressing topics, a risk protocol was in place with clear pathways for managing participant distress in the interviews. Following the interviews, participants were contacted over the phone within three days after the interview to check in due to the distressing topics raised in the discussion.

### 2.3. Data Collection

A topic guide was used to provide a basic structure to the interviews whilst allowing for flexibility to expand the discussion. (see Appendix A) A patient and public involvement (PPI) group provided input into topic guide development due to the sensitive nature of the topic. A total of 17 online, one-to-one, semi-structured interviews were conducted and recorded over Zoom. All interviews were carried out by one researcher and transcribed manually and verbatim. All data were anonymised to ensure confidentiality.

The researcher kept a journal of their thoughts throughout the study for reflection. Recruitment was stopped once data saturation was reached.

### 2.4. Patient and Public Involvement Statement

A PPI group consisting of seven medical students in their clinical years, both male and female, was formed to provide alternative perspective and unbiased opinions throughout the study, thus supporting the FSCT approach. The group was debriefed and their opinions on the topic guide and participant information leaflet explored. The topic guide was adjusted accordingly based on the feedback from the PPI group. The group shared their thoughts on the findings of the study ensuring a neutral conclusion based on the data collected.

### 2.5. Data Analysis

NVivo software was used to generate and refine codes and create themes. Inductive thematic analysis was used to analyse the data generated. Randomly selected sets of anonymised data were coded independently by external researchers. The codes and analysis were then compared the anonymised data sets to allow for data triangulation, thus increasing the validity of the research. There were no disagreements with data analysis that required resolution. Furthermore, member validation was employed to improve credibility.

## 3. Results

There was a diverse range of participant demographics. A summary table of these demographics can be seen in Table 2.

Following the completion of data analysis, four key themes (Table 3) arose from the data set. 

### 3.1. Theme 1: Experiences of Sexism

Participants shared encounters of sexism during their clinical placement. Final year students faced more encounters of sexism than younger students, likely due to longer exposure while on clinical placement years. Patients were the major perpetrators of physical sexual harassment, whereas senior doctors more frequently employed stereotyping and microaggressions. Surgical specialties were noted to be the most discriminatory.

#### 3.1.1. Sexual Harassment

Sexual harassment experienced by medical students was both physical and verbal, with perpetrators being trainee doctors and patients.

Participants reported incidences of physical sexual harassment from patients whilst on wards taking patient histories and carrying out examinations.


*“I have had my bum pinched once after taking someone’s bloods.”-P0003*


Verbal comments of a sexual nature were also encountered from patients. Comments occurred in one-to-one scenarios as well as in front of senior medical staff. Participants struggled with such encounters and hoped that the presence of senior medical staff would provide them with support and reassurance. However, participants were often left unsupported when comments went unacknowledged and unchallenged.


*“The elderly gentleman just pointed at my breasts and goes, “Look! Look at the size of those”. I didn’t know how to react. And all the staff just looked at me, didn’t say anything, and continued the consultation.”-P0016*


Other experiences of inappropriate sexual intentions occurred with doctors. Attempts were made by perpetrators to contact and meet students outside of placement. One participant shared an experience of a doctor stalking their social media and attempting to contact them.


*“We didn’t exchange names or anything like that. So then in the evening, about 11 o’clock at night, he added me on Facebook. Presumably he got my name from my badge.”-P0014*


Participants were placed in uncompromising positions by superiors. Several incidents were discussed where doctors abused their position of power. Some students were forced to choose between accepting advances from staff or compromising their learning. 


*“he’s like, Yeah, I’ll sign it off if you go out with me.”-P0002*


#### 3.1.2. Negative Teaching Experiences

The majority of participants indicated that teaching staff privileged male students by focussing their attention on them during teaching. Experiences varied from subtle microaggressions, including body language such as avoiding eye contact and facing towards the males, to more overt experiences of ignoring female students in conversation.


*“He just didn’t want to look at me, it was just so awkward.”-P0011*



*“He would speak to literally the boys and not speak to me.”-P0010*


Occasionally, direct comments of discrimination towards the female students were made. Some staff would explicitly refuse to teach female students based on stereotypical assumptions of females having families later in life.


*“They literally said to me “what is the point of me teaching you when you’re a woman and you’re just going to choose your babies over your career and want to be a GP anyway?”.”-P0013*


#### 3.1.3. Gendered Stereotyping

Labelling females as nurses appeared to be a frequent occurrence. Patients would often assume males to hold the role of doctor and females the role of nurse. Students value and respect nursing as a career; however, they also want to be addressed correctly based on their skill set and expertise.


*“I think nurses do an amazing job. And I’d be proud to be a nurse. But I worked really, really hard to get here. And I keep working hard. And it’s just frustrating when I get called nurse.”-P0007*


Staff of all grades made assumptions on female students and motherhood, including the desire and ability to have children and be the stay-at-home parent. 


*“They just look at you and all they see is a carrier of children.”-P0006*


Unsolicited advice was frequently encountered based on such assumptions and stereotypes. 

Advice regarding careers was aimed solely at female medical students. Comments were unprompted, uninvited, and focused on choosing specialties to fit into family life. Participants were mostly discouraged from surgical specialties, whilst GP was considered suitable for females.


*“You get told that that specialty is not for you because you need to have children”-P0003*



*“I’ve been told that I should be a GP because it means that you can have a family”-P0008*


Differences in experiences between White and Asian participants were identified, where intersectional accounts, including cultural stereotypes and sex, arose as a minor theme. Students, particularly those from Asian backgrounds, faced assumptions of interest in a GP career due to their ethnicity and cultural importance of family.

#### 3.1.4. Microaggressions

Several students experienced patronising and condescending behaviour from male peers. Simple concepts would often be “mansplained” although it was unsolicited, causing frustration.


*“Things that annoy me the most are when other medical students patronise me, because it happens a fair bit…it’s annoying that he would even think to say that when I’m sure he wouldn’t say that to a guy.”-P0001*


Participants additionally noted the condescending use of pet names for female students. Such terms appeared undermining and unprofessional in the workplace, especially when used by senior staff.


*“You don’t say sweetheart to the boys. Why? Because it feels very patronising.”-P0006*


Alongside the numerous experiences of negative sex discrimination towards females, a minority of participants experienced a patriarchal patronising behaviour. Overcompensation occurred from some seniors highlighting that they are much better than males. Students often felt uncomfortable and were led to question the root cause for such excessive behaviour.


*“It was always a male consultant overcompensating, saying that all of the female consultants were fantastic and don’t get me wrong. It’s nicer to hear something good than something terrible, but I sort of got the impression, it was a bit much, you know, almost like they were trying to prove a point”-P0003*


### 3.2. Theme 2: Impact of Sexism

The analysis found that sexism impacted students in three principal ways: emotional and psychological impacts, development and learning, and “Self-consciousness”.

#### 3.2.1. Emotional and Psychological Impacts

Although experiences varied, there appeared to be a common negative psychological impact. Experiences are subjective and the effects can vary between individuals; however, a range of negative feelings were described by all following the encounters. Many felt hopeless about the situation and the career they were embarking on.


*“It’s really disheartening”-P0004*


There was a general belief that succeeding in medicine required participants to give up all other interests. Many felt they had to choose between a medical career or other interests, having to sacrifice one as both were not possible.


*“It makes it feel like the career that I want is just so out of reach just because of what I want outside of medicine”-P0010*


Emotions varied from shock and anger to pure sadness. The experiences accumulated over time, leading to an overwhelming emotional impact, in turn affecting participants’ focus and engagement with learning.


*“When I first was discovering how sexist medicine was, it got me so down and depressed, it made me really disengaged with the whole thing”-P0011*


Participants concluded that such incidents felt worse when occurring in the workplace as such an environment is meant to be safe to learn and work.


*“It’s uncomfortable. Especially that it’s supposed to be a professional environment.”-P0005*


Experiences from staff negatively influenced specialty interest and resulted in students dismissing particular specialty choices. The frequency of such encounters only reinforced these decisions. 


*“The decisions I’ve made for example not wanting to do surgery. It’s not based on that one session that I had, but it’s experiences like that, that happen over and over.”-P0017*


Rumination of negative experiences was another impact of sexism. In some cases, participants spent weeks to months thinking about encounters, compounded further by the inability to process in a safe space and lack of resolution. This preoccupation took away from time spent studying.


*“A passing comment it can really sit with you, and I’ll think about it for weeks after. Like I still think about it now and that was in July last year.”-P0004*


#### 3.2.2. Impact on Learning

A multitude of actions taken by students to deal with the encounters consequently affected their learning. Avoidance was a behaviour executed by the majority of students following their experiences. Students avoided patient contact as well as particular areas of the hospital or specific wards. This in turn reduced their clinical exposure, leading to a detrimental impact on their learning.


*“I never went back on the ward. That’s how I dealt with it.”-P0002*


Some responses by avoidance were even more extreme with one student avoiding the entire region by changing their foundation year application in order to eliminate any possibility of encountering the perpetrator in the future.


*“That made me not really want to go to clinics anymore, and it’s changed where I was going to apply for my foundation year programme.”-P0013*


Students also missed out on future teaching sessions when the perpetrators of sexism were staff, which resulted in many missed educational opportunities.


*“I was then missing the teachings so I was missing out on what I could have been taught.”-P0010*


The impact was considered greater when the perpetrators were staff, as professionalism was expected from those in positions of power.


*“When it comes from a teacher, a mentor, it affects me more”-P0012*


Encounters were recognised to have an impact on education. Following the event, students found themselves struggling to maintain focus, thus having a detrimental impact on their productivity.


*“I literally, I couldn’t concentrate on anything!”-P0006*


Students engaged less in teaching sessions with perpetrators, further reducing their ability to learn and develop knowledge. Engagement was negatively impacted in various ways with some less willing to attend sessions and others less able to listen and actively participate in lessons.


*“You might be less likely to go, you might be less likely to listen, to trust them.”-P0003*


Education was also impacted with students doubting their abilities and undermining their confidence. Lack of confidence and feelings of insecurity led to reduced interaction with studies.


*“It can feel like a confidence knock”-P0012*


#### 3.2.3. Conscious Behavioural Changes

Participants were also impacted directly by having to consider and plan their outfits carefully to diminish any risk of sexualisation.


*“It makes you more wary around what clothes I can wear.”-P0016*


As well as outfit planning, one participant was more cautious about visibility of their name badge and who they introduced themselves to following an uncomfortable experience of harassment over social media.


*“I do feel a bit more protective over my name badge now. So I’ll try and cover it with my hair, or I won’t wear it as often.”-P0014*


### 3.3. Theme 3: Barriers and Facilitators to Reporting

Barriers and facilitators to reporting were a major theme. The potential facilitators to reporting, identified by participants, were the reverse of the stated barriers, illustrated below.

#### 3.3.1. Hierarchy and Power Imbalance

One of the most common barriers to reporting was hierarchy. The power imbalance between senior staff perpetrators and students was considered to be a key deterrent. Participants struggled to challenge sexist perpetrators in positions of power and did not feel empowered to report and whistleblow out of fear of reprisal. 


*“We shouldn’t, we can’t complain because he’s so much more senior than us.”-P0004*


#### 3.3.2. Negative Experiences and Futility of Reporting

Of the 17 participants interviewed, 16 did not know the official reporting procedures within medical school. This lack of knowledge on reporting processes prevented students from reporting sexist experiences.


*“I’ve thought countless times shall I report them? But then I don’t know how.”-P0012*



*“I don’t actually know what the official reporting procedures are.”-P0009*


Previous futile encounters of unofficial reporting within medical school have resulted in students lacking faith in the reporting process, thus being less willing to report future encounters.

Those that had previously reported incidents felt unsupported by those in positions of power. Reported incidents against male peers were dismissed and not investigated, thereby perpetuating injustices experienced by victims. Participants felt that the medical school was complicit in creating cultures which provided a “playground for men’s sexist behaviours”.


*“[Medical School Management] said ‘the goal of medical school is to provide a safe environment for men to make these mistakes so that when they qualify as a doctor, they don’t go on to make those mistakes… and lose their registration and harm patients’.”*
*-*
*P0011*


There was a disconnect between medical students and seniors within medical school. Students did not feel their concerns were taken seriously or that the necessary outcomes were achieved. 


*“Barriers to reporting aren’t necessarily that the roots aren’t there, but the people that you’re reporting to don’t take it seriously.”-P0015*


Some students never heard back after contacting medical school to report an encounter.


*“I got an email from the uni being like, ‘we’ve seen your thing and we’re happy you brought this up to my attention. We’re going to make a forthcoming meeting with you to see what’s happened’. No one ever made a meeting. That was 11 weeks ago now.”-P0004*


When reports were investigated, females found that the male seniors’ testimony was favoured over theirs.


*“They took his word over mine.”-P0016*


Experiences of sexism without recourse to physical evidence led to students becoming disenfranchised with the process. There was a common recognition of most encounters being impossible to prove, resulting in “one word person’s word against another”.


*“You have to prove something that’s happened, and a lot of things are very difficult to prove, because there is no objective proof you can provide.”-P0013*


As well as complaints being dismissed, some were set aside and not passed on, resulting in no appropriate action. Instead, the problem was focused back on the victim, forcing them to deal with the consequences.


*“Wellbeing got in touch with me saying, “we’re sorry this happened. We don’t know about feeding it back, but we can certainly do sessions to deal with your anxiety and help you”.”-P0011*


Participants believed that the reporting process was time consuming, leading to under-reporting. With encounters being so frequent and reporting processes being so long and exhausting, students felt they did not have the energy to report individuals whilst also balancing their studies. This issue was further extenuated by other barriers, including undesirable outcomes from past experiences.


*“It’s not really worth putting all that effort into reporting it and not getting anywhere with it.”-P0014*


Confusion with the threshold to warrant reporting was another barrier. Students were unsure of which encounters were considered serious enough to report as they had not been provided with teaching on this matter.


*“How bad does the sexism have to be for something to be done about it?”-P0007*


Another barrier that was made evident by multiple participants was the fear of their learning being impacted. Concerns about perpetrator staff finding out and then bullying particular individuals, leading to repercussions for the students’ education, were observed.


*“I don’t want to put myself at risk of anything bad happening, by reporting this.”-P0006*


### 3.4. Theme 4: Student Recommendations

#### 3.4.1. Raising Awareness and Empowering Students

The over-arching sub-theme under recommendations was increasing education. Students wanted teaching on defining and recognising sexism as well as education on reporting.


*“I think there should be some formal teaching at uni about what sexism can look like in the medical sphere, what they would like you to do about it. I think if uni were giving us a lecture saying- this is the reporting process, this is what it can look like, we will endorse and we will back you if you were to complain about it.”-P0011*


More education encompassing various forms of teaching methods as well as a range of topics were desired. Clinical communication sessions were considered, by participants, to be the most appropriate style of teaching to practise managing sexism in different scenarios, including responding to patients as well as calling out staff in a polite and professional manner.


*“I think we could do with like a couple of communication sessions on how to deal with these things at the time, how to recognise them, we need a better reporting process in place and where to access the reporting system as well.”-P0014*


The importance of improving awareness and education of the reporting procedures was highlighted alongside the importance of responding and dealing with encounters.


*“Medical school teaching us how to respond to issues and teaching us how to be able to report them would really help.”-P0012*


Participants mentioned that education on sexism should target everyone from students in medical school to hospital staff in order to educate the perpetrators. It was emphasised that the responsibility to change should not be with the victims, but instead with the perpetrators.


*“We don’t really teach men how to not be sexist.”-P0011*


#### 3.4.2. Improving Reporting Outcomes, Recognising the Severity of the Issue and Encouraging Reporting

Another suggestion from students included improving outcomes of reports. Such improvements would entail seniors within placements and medical schools to recognise the detrimental impacts of sexism on students and their education and overseeing such pressing matters with care and seriousness.


*“I guess the med school and hospitals could take the issue more seriously.”-P0015*


Participants stated that reporting could be facilitated and encouraged by providing an anonymous reporting system that would not negatively impact those reporting.


*“I think each trust should have a way of reporting sexism or sex discrimination…in an anonymous way.”-P0009*


#### 3.4.3. Provision of a Safe Space for Discussion

Aside from providing education and improving reporting outcomes, a common theme of providing a safe space to discuss experiences emerged. The provision of a safe space to share encounters and feelings would support the victims as well as raise awareness of the issue and its frequency.


*“More education earlier on in the degree, a harder stance against these issues and providing forums for people to share their experiences.”-P0017*


## 4. Discussion

This study reports the experiences and impact of sexism on female medical student participants while on clinical placements; perpetrators, all male, included staff, students and patients. Experiences varied between students and included sexual harassment, microaggressions and gendered stereotyping. The impacts following such encounters were detrimental to students’ mental health and their educational journey. Experiences of sexism affected students’ ability, confidence and motivation to engage with clinical placements, particularly in some specialties to avoid perpetrators, often consultants, thus disrupting learning opportunities. 

Students also reported multiple barriers to reporting, lack of visibility and understanding of reporting procedures, futility in reporting due to a lack of trust in systems attributable to prior or observed negative reporting experiences and a fear of repercussions on their education. Trauma from the experience and fear could also act as additional barriers, increasing the feeling of alienation and isolation, among other impacts.

Institutional leadership was experienced as passive, with complainants left to manage the impact and pick up the emotional pieces. Similar challenges to reporting have been evidenced previously [35], including powerlessness, a lack of transparency, and fear that whistleblowing would have negative consequences on their learning. These challenges left complainants feeling invalidated with no recourse to justice or closure coupled by institutional apathy or failure in holding perpetrators to account; such cultural and systemic barriers perpetuate inequality within medicine [36,37,38,39].

Negative experiences of formal disclosure contribute to the under-reporting, undermining transparent processes for monitoring incidences and outcomes. Parallels can be drawn between NHS staff whistleblowing experiences and students reporting sexism, suggesting an overlap in difficulties faced when reporting to institutions [40,41]. 

Existing evidence highlights inter-generational differences in attitudes and cultures with women having to “put up with” rather than resist, challenge and change the status quo [42,43,44]. Such views have developed from years of patriarchy imposed upon society [45]. The institution holds power over the students and by not taking appropriate steps following an event, change is prevented. Students are opposing and resisting these dominant discourses by continuously reporting incidents and voicing their needs for appropriate outcomes.

Universities hold responsibility for student wellbeing whilst healthcare providers are responsible for clinical exposure provision [46,47]. Uncertainties over responsibilities for students’ experiences on clinical placement and remits in investigation processes lead to a lack of justice and accountability for perpetrators’ actions.

Encounters of sexism and their aftermath have detrimental impacts on performance and can lead to a decrease in job satisfaction, which in turn negatively impacts mental health [48,49,50,51]. Institutions and staff have a duty of care to provide a positive and safe working environment for all students. 

Other impacts included missed learning opportunities causing gaps in knowledge as students avoided certain specialties and experienced reduced engagement. Discouraging females from entering specific specialties also leads to a lack of diversity within teams and areas of medicine, while perpetuating discriminatory workplace cultures. Diversity is vital for strong functioning teams and is known for better productivity, efficiency and outcomes [52].

To tackle sex inequalities, the issue needs to be recognised by students and institutions alike [53,54]. It is necessary to acknowledge the disadvantages faced by women and the privileges afforded to hegemonic men due to sexism [53]. The recognition of such inequalities is needed to support women in challenging sexism and demand institutional change. Training is needed for all students to recognise bystander complicity and address discrimination by upstanding, taking ownership and adopting active allyship to those facing discrimination in order to build a zero tolerance culture [17,55,56].

Awareness and communication skills are required by all in managing such encounters. All students should be equipped in assertiveness skills to deal with discrimination, sexual harassment, and bullying. Students are requesting education on the topic, showing that it is not only needed but also strongly desired [57,58]. Varied teaching methods were suggested to allow for careful integration into the curriculum and maximise engagement. These included lectures, small group teaching sessions as well as clinical communication scenarios with actors. Such opportunities provide safe spaces for students to rehearse handling sexist perpetrators in a professional manner [59,60]. Creating a safe space for discussion also allows for victims to share experiences and emotionally process, thus supporting mental health.

Students across other universities are experiencing and speaking out about similar encounters; this was evidenced when victims from other institutions reached out during this project [61]. Sexism and discrimination generally need to be actively monitored in order to provide future evidence. Partaking in research has the potential to empower and provides agency to participants, can serve as a source of catharsis and has the potential to impact social change. Steps need to be taken to understand how to bring about change and improve student experiences nationwide. Representing (i.e., BMA) and regulatory (i.e., GMC) bodies need to be more proactive in ensuring that medical students are protected and provided with safe learning environments, ensuring medical schools are proactive in their commitments to deliver this.

### 4.1. Reflexivity

Reflexivity is integral to feminist-informed qualitative research as it reflects on the positionality and power of the researcher and participant while recognising emotionality as a valuable data source [62]. Effective self-reflection is essential to mitigate personal bias while recognising the insights it provides. The researcher was a novice interviewer and an undergraduate female medical student at the university. As a female medical student who experienced sexism on placement, the researcher recognised the risk for potential bias and the subconscious assumptions that they may hold but also the value in using their experiences to generate alliance, in offering empathy and in the interpretative process while avoiding over-identification. Feminist research aims to reduce power imbalance. Thus, a participatory model was adopted to share stories and encourage reciprocity to remove hierarchy and develop rapport [63]. 

Triangulation, member validation, transparency and reflexivity enhanced research credibility.

### 4.2. Strengths and Limitations

No previous qualitative research has explored the experience and impact of sexism on medical students and barriers to reporting. 

Purposive sampling and snowball sampling were used to maximise variation. The participant demographics were diverse and transferable to other institutions within England.

Data saturation was reached with 17 participants.

Semi-structured one-to-one interviews allowed for the flexibility needed for an in-depth discussion.

Interviews were conducted by one novice researcher to mitigate their limited experience; they were supervised by an experienced qualitative supervisor who provided feedback on interview transcripts and facilitated reflective supervision to enable the novice researcher to improve their interviewing and research skills.

One researcher is a female medical student who has experienced sexism on placement, thus their experience may have influenced their data collection and analysis. While also providing insight, reflexive practice through journaling and discussions with the supervisor were employed throughout to counterbalance potential biases and over-identification. The researcher reflected on their feelings after the interviews and during the transcription process to minimise the likelihood of bias. Semi-structured interviews also allowed for the interviewer to ensure that the questions were objective to minimize leading interviewees and reduce personal bias. Frequent discussions with an experienced supervisor further aided reflexivity.

## 5. Conclusions

Urgent change is required to address sexism in medicine, including at an undergraduate level. It is necessary to address sexism, discrimination and other drivers of continued inequity in medicine including gender bias. National measures by the General Medical Council are needed to regulate and address sexism in the medical curricular. Appropriate action needs to be taken to tackle attitudes and behaviours which disempower and undermine the contribution of current and future female doctors. 

Sexism within medicine is widely evidenced by medical students and doctors in clinical practice to negatively impact well-being, mental health and general training and development. To address this issue, it is essential to recognise and address the barriers that impede the reporting of such incidents, and reform the current systems, policies, and processes to manage and provide justice to those affected while offering psychological safety and support. Students need to be trained and empowered to respond effectively, but attitudes and practices that contribute to the perpetuation of sexism in the medical field must be challenged and changed to create an inclusive, equitable and safe environment for education and training for all medical students.

## Figures and Tables

**Figure 1 healthcare-11-01002-f001:**
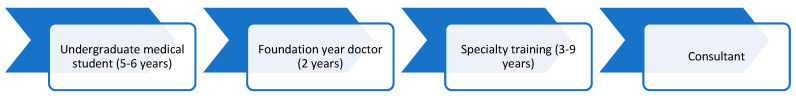
Training pathway in England [12].

**Table 1 healthcare-11-01002-t001:** Participant Inclusion Criteria.

English speaking
Woman or non-binary
University of X medical student
Clinical years (3–5)
Have experienced sexism on placement
Access to device with stable internet connection to carry out interview

**Table 2 healthcare-11-01002-t002:** Participant Demographic Table.

Demographic	Category	Number
Gender	Woman	17
Ethnicity	Asian Indian	4
	Mixed Asian + White	2
	White British	11
Disability	No	17
Year group	3	4
	4	4
	5	9
Graduate entry	No	14
	Yes	3
Religion	Christian	3
	Hindu	3
	Muslim	1
	No religion	10
Sexual orientation	Straight	12
	Gay	1
	Bisexual	3
	Other	1

**Table 3 healthcare-11-01002-t003:** Summary of Major themes and sub-themes.

Major Themes	Sub-Themes
Experiences of sexism	Sexual harassment
	Negative teaching experiences
	Gendered stereotyping
	Microaggressions
2. Impact of sexism	Emotional and psychological impacts
	Impact on learning
	Conscious behavioural changes
3. Barriers to reporting	Hierarchy and power imbalance
	Negative experiences and futility of reporting
4. Student recommendations	Raising awareness and empowering students
	Improving reporting outcomes, recognising the severity of the issue and encouraging reporting
	Provision of safe space for discussion

## Data Availability

Not applicable.

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
