# Peer review of "Female Medical Students’ Experiences of Sexism during Clinical Placements: A Qualitative Study"

_healthcare, 2023, doi:10.3390/healthcare11071002_

Round 1

Reviewer 1 Report

Thank you for submitting the manuscript to the journal "Healthcare." The topic is highly fundamental and helpful in improving the actual situation of sexual discrimination and harassment.

According to the COREQ checklist for the quality of qualitative research articles, please check the following issues.

- Normally demographics of study participants come at the beginning of results. You should mention how you asked and selected study participants in the method section and move the demographics to the results. Especially the order of Tables 1 and 2 should be replaced with each other because the inclusion criterion is the entrance to start the study, and demographics is the exit of it.

- In 2.3, you explain that one interviewer carried out all interviews. Please provide detailed information about the interviewer, including gender, generation or age, healthcare qualification, how much interviewing training or experience the interviewer has, and the relationship between the author and the interviewer (if they are only one person, that is also fine). 

- How many students were recruited, and how much was the acceptance rate? If there are specific reasons for rejection, please describe them.

This should be a significant manuscript. So, I'll wait for the re-submission of this manuscript again.

Author Response

We would like to thank reviewer 1 for their time to review and we greatly appreciate the comments.

Please see attached the reviewer comments and our responses.

Reviewer 2 Report

Thank you very much for giving me the opportunity to read your manuscript. This is very important content to be discussing in our current landscape and provides ample opportunity to change attitudes and inspire future research. To help amplify those opportunities, I have a few suggestions which could strengthen this manuscript.

p. 2, Line 74-76: This sentence reads a little clunky and needs either some punctuation or split up.

p. 2, Line 78: While this study does provide good insight, framing the understanding gained from findings as "comprehensive" seems ambitious. Given the sampled students are from one medical school per the inclusion criteria and represent just over 1 percent of the female population of that school based on the stats listed, the generalizability of the findings could be low.

p. 3, Lines 113-118: This could just be from the peer review copy but verify the fonts in this section.

p. 5-6: Headings for Negative Teaching Experiences, Gendered Stereotyping, and Microaggressions need to be fixed to 3.1.2, 3.1.3, and 3.1.4.

p. 6, Lines 216-217: This sentence discusses the differences between multiple ethnic groups that were found, but the demographics listed in Table 1 only listed White, Asian, and Mixed Asian/White as demographics among interview subjects. Why are other ethnic groups listed here?

p. 7, Lines 263-278: These findings, while there are definite impacts to fall under heading 3.2.1, seem to be more focused on learning and educational outcomes and might fit better under heading 3.2.2. 

p. 8: Subheadings 3.3.2 and 3.3.3 feel like the findings could fit under the Negative Experiences and Futility of Reporting in 3.3.4 and may not have the support to be their own subheadings.

p. 9: The findings under subheading 3.3.4 are supposed to illustrate the Negative Experiences and Futility of Reporting. Subheading 3.3.2 said that 16 of 17 interview participants did not know reporting procedures, but in this section, 7 respondents give their experiences after reporting. I'm confused as to the disconnect. Are the reporting procedures in 3.3.2 more formal in nature while 3.3.4 is informal? If so, this needs to be explained more clearly.

p. 10: Subheadings 3.4.2 and 3.4.3 could be combined since they are both related to increasing reporting of sexism issues.

p. 12, Lines 523-527: While I respect the efforts have been made to try and eliminate the potential bias in this study, I think more on how those efforts were conducted and minimized would help in this section.

Author Response

We would like to thank reviewer 2 for their time to review and we greatly appreciate the comments.

Please see attached the comments and our responses.

Round 2

Reviewer 2 Report

Thank you for addressing the comments provided regarding the first version of this manuscript. Changes helped to improve flow, better explain the findings, and provide a more complete paper for readers. Excellent work.